META-RESEARCH ARTICLE

# Explosion of formulaic research articles, including inappropriate study designs and false discoveries, based on the NHANES US national health database

Tulsi Suchak[1], Anietie E. Aliu[1], Charlie Harrison[2], Reyer Zwiggelaar[2], Nophar Geifman[1], Matt Spick [1]*

**1** School of Health Sciences, Faculty of Health and Medical Sciences, University of Surrey, Guildford, United Kingdom, **2** Department of Computer Science, Aberystwyth University, Ceredigion, United Kingdom

* matt.spick@surrey.ac.uk

## Abstract

With the growth of artificial intelligence (AI)-ready datasets such as the National Health and Nutrition Examination Survey (NHANES), new opportunities for data-driven research are being created, but also generating risks of data exploitation by paper mills. In this work, we focus on two areas of potential concern for AI-supported research efforts. First, we describe the production of large numbers of formulaic single-factor analyses, relating single predictors to specific health conditions, where multifactorial approaches would be more appropriate. Employing AI-supported single-factor approaches removes context from research, fails to capture interactions, avoids false discovery correction, and is an approach that can easily be adopted by paper mills. Second, we identify risks of selective data usage, such as analyzing limited date ranges or cohort subsets without clear justification, suggestive of data dredging, and post-hoc hypothesis formation. Using a systematic literature search for single-factor analyses, we identified 341 NHANES-derived research papers published over the past decade, each proposing an association between a predictor and a health condition from the wide range contained within NHANES. We found evidence that research failed to take account of multifactorial relationships, that manuscripts did not account for the risks of false discoveries, and that researchers selectively extracted data from NHANES rather than utilizing the full range of data available. Given the explosion of AI-assisted productivity in published manuscripts (the systematic search strategy used here identified an average of 4 papers per annum from 2014 to 2021, but 190 in 2024–9 October alone), we highlight a set of best practices to address these concerns, aimed at researchers, data controllers, publishers, and peer reviewers, to encourage improved statistical practices and mitigate the risks of paper mills using AI-assisted workflows to introduce low-quality manuscripts to the scientific literature.

**Data availability statement:** The extracted dataset used in this work is included in full within S1 Data, Table A: Metadata for research papers extracted by the search strategy. The Supporting information additionally includes the data underlying each of the Figures presented in this work. All code libraries were used without modification or customization.

**Funding:** CH and RZ were supported by the Biotechnology and Biological Sciences Research Council (BB/Y006933/1). The funders had no role in study design, data collection and analysis, decision to publish, or preparation of the manuscript.

**Competing interests:** The authors have declared that no competing interests exist.

**Abbreviations:** AI, artificial intelligence; API, Application Programming Interface; Bridge2AI, Bridge to Artificial Intelligence; COPE, Committee on Publication Ethics; CRP, C-reactive protein; NHANES, National Health and Nutrition Examination Survey; SII, systematic immune-inflammation index.

# 1. Introduction

The quantity of biological data available to researchers has increased dramatically in recent years, leading to more opportunities for data-driven research. As more information becomes available in artificial intelligence (AI)-ready formats, research—when performed in line with best practices—should become faster and more reproducible. The wide availability of such datasets can, however, introduce new problems, by facilitating end-to-end AI-supported manuscript production on a large scale. This is a practice which may be adopted by paper mills, defined by the United2Act Research Working Group as covert organizations that provide low-quality or fabricated manuscripts to paying clients [1].

Here, we systematically investigate research papers which used data extracted from the National Health and Nutrition Examination Survey (NHANES) [2], a cross-sectional data source originally established to assess the health and nutritional status of adults and children in the United States. NHANES combines interviews, physical examinations and laboratory tests to collect comprehensive data on the prevalence of diseases, risk factors, and health trends. NHANES surveys are conducted on a two-year cycle and aim to recruit around 10,000 participants per survey. While many variables are collected on a continuous basis, others have been included or excluded at different points, as areas of interest to stakeholders have changed, and participants are freshly recruited for each survey. The most recent NHANES survey for which data are available (covering 2021–2023) included over 700 variables. In terms of accessibility, NHANES is an AI-ready dataset, in line with the criteria set out by the NIH Bridge to Artificial Intelligence (Bridge2AI) Standards Working Group [3].

The long-standing nature of NHANES has led to the creation of R and Python libraries which provide *inter alia* automated search, extraction, and analytical tools, providing standardized workflows and improving reproducibility (Table 1). Such tools, alongside other widely used coding environments and libraries, can aid significantly in the rapid production of results and subsequent publications. General purpose NHANES libraries are tabulated below (specific libraries also exist for individual conditions, for example for data relating to asthma, but these do not provide the breadth of access required for the analyses investigated here). The ability of researchers to automate the data extraction process via an Application Programming Interface (API ; consistent with FAIR guidelines that data be retrievable by identifier using a

**Table 1. R and Python libraries for the standardization of data extraction and analysis from NHANES.**

| Library | Environment | Most recent release |
|---|---|---|
| nhanesA [4] | R | 1.1 (2024) |
| RNHANES [5] | R | 1.1.0 (2019) |
| NHANES pyTOOL [6] | Python | 1.0 (2023) |
| nhanes-dl [7] | Python | 0.0.18 (2023) |
| Pynhanes [8] | Python | 0.0.20 (2024) |

standardized communications protocol), allows for the transfer of data directly to machine learning environments, facilitating rapid and comprehensive data exploration.

The ability to extract data via an API directly into machine learning environments such as R or Python can transform productivity, with the number of hypotheses that can be tested constrained only by computational access, but this can also carry risks. A focus on single-factor analyses can be especially problematic, given the multifactorial nature of many illnesses, as well as the challenge of differentiating between predictors that are specific to a health condition versus those shared across different disease types [9]. In addition, the ability to generate large numbers of machine-learning models allows for rapid post-hoc investigation of alternative hypotheses, should the main a priori hypothesis not be supported (a form of hypothesizing after the results are known, or HARKing) [10–12]. With ready computational access, it is possible to conduct a broad search for any combination of indicator, health condition, cohort, and time window that yields a low p-value. While data dredging is a well-described phenomenon [13–16], direct-to-AI pipelines can make formulaic research pipelines more productive than has previously been possible. This productivity gain is likely to be particularly attractive to paper mills.

In this work, we conducted a systematic literature search over the last 10 years to retrieve potentially formulaic papers analyzing NHANES data, and analyzed these manuscripts for common themes around statistical approaches, study design, or results that were not translational in nature. We also aimed to identify whether these issues provided a case study of the risks of AI-supported workflows being adopted by paper mills, from workflows to automate data dredging and machine learning, to manuscript preparation using generative AI.

## 2. Results

### 2.1 Systematic identification of articles: Bibliometric analysis

The results of the systematic search strategy to identify NHANES-derived associative research papers published over the past decade are shown in Fig 1.

The 341 identified reports were published across a number of different journals (147 journals in total); all articles were successfully retrieved. The top 10 journals accounted for 43% of the articles. This is also shown in a tree map format in Fig 2 below, with the full data in S1 Data Table A. The average impact factor of the journals publishing these papers was 3.6. Three journal families accounted for over half of the manuscripts identified in this review: Frontiers Media SA (22%), BioMed Central Ltd (18%) and Springer (13%).

In terms of trends over time, an average of 4 single-factor manuscripts identified by the search strategy were published per year between 2014 and 2021, increasing rapidly from 2022, with 190 in 2024 up to 9 October. There has also been a general increase in health data-driven research (Fig 3B shows this trend for the search term 'biobank' as a simple example), but this wider growth does not explain the scale of the increase seen here. There was also a change in the origin of the published research. From 2014 to 2020, 2 out of 25 manuscripts had a primary author affiliation in China, compared with 292 out of 316 manuscripts between 2021 and 2024.

### 2.2 Statistical issues arising from single-factor study design

**2.2.1 Identification of multifactorial conditions analyzed as single-factor problems.** One hundred sixty-nine predictor variables were proposed as having statistically significant associations with the different health conditions reviewed in this meta-analysis. The three most commonly cited independent variables were the systematic immune-inflammation index, sleep health and serum vitamin D concentrations. The network analysis in Fig 4 illustrates the number of identified relationships and the complexity of the interactions, for the 16 health conditions in this systematic review with 4 or more associated predictors. The data underlying Fig 4 are aso shown in full in S1 Data Table B.

**2.2.2 False discovery correction.** For the 138 health conditions studied in this meta-analysis, the number of studies proposing an association with an independent variable (biomarker or health/clinical indicator) ranged from 1 to 28; the

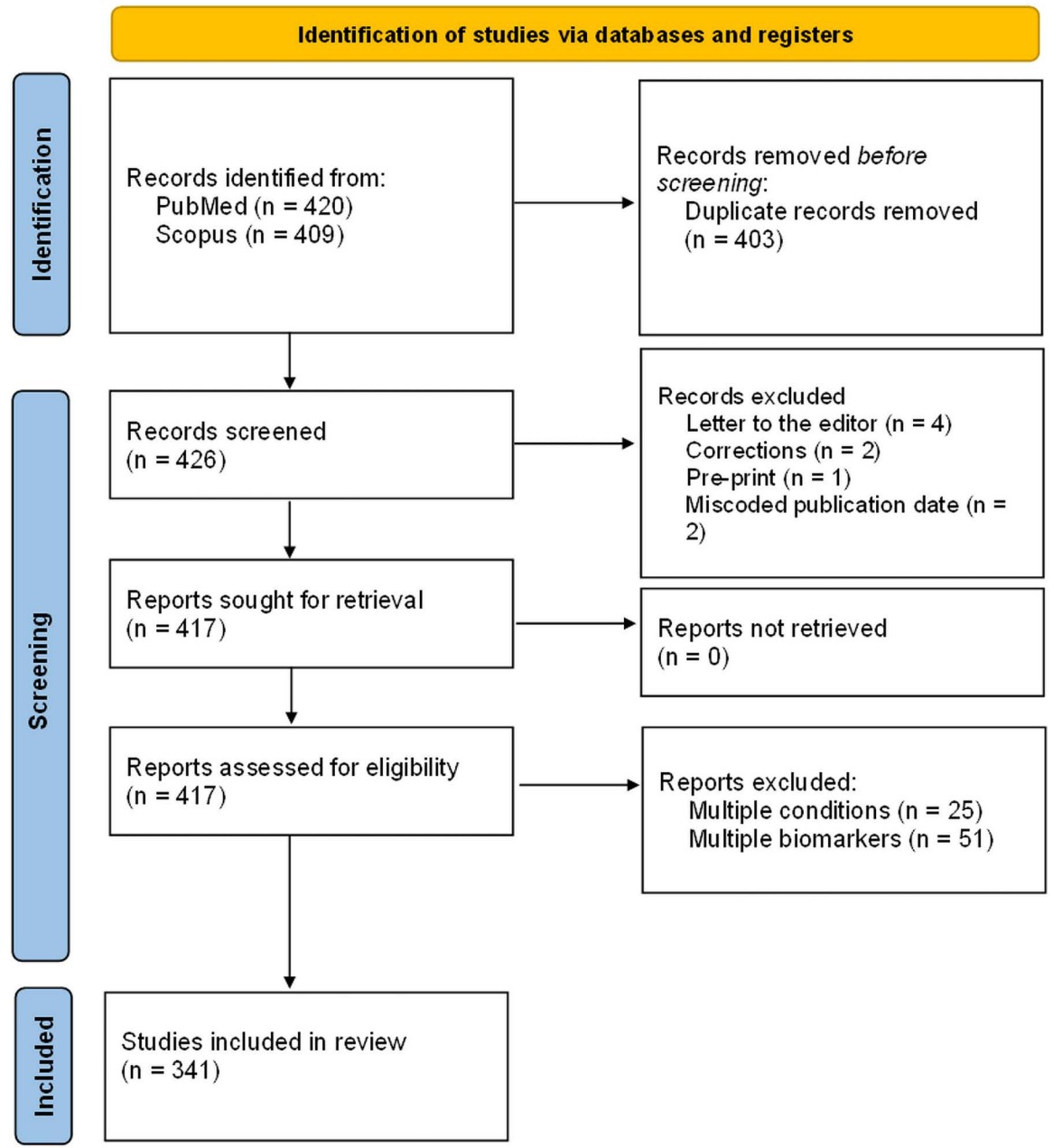

**Fig 1. PRISMA four-phase flow diagram of the article identification process.**

mean average was 2 studies per condition, but the distribution was skewed, and most conditions were only associated to one independent variable. Some variables were included as both predictors and as outcomes, for example, C-reactive protein (CRP) was investigated as being associated with periodontitis (PMID 37481511), but in another study, a health behavior index was investigated as a predictor of elevated CRP levels (PMID 38628050). In the most extreme case, additional manuscripts could be generated by simply reversing dependent and independent variables in any statistical analysis, increasing the number of combinations of predictors and outcomes with no physiological justification or

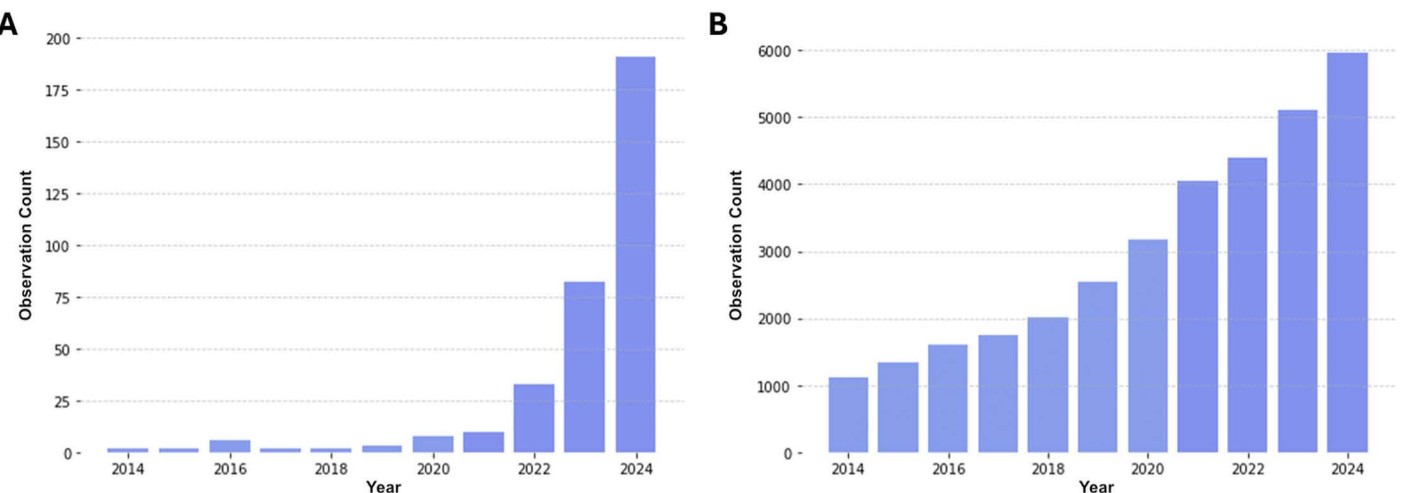

**Fig 2. Summary of distribution of journals: Tree map of 341 manuscripts by journal, grouped by publisher family.** BioMed Central is part of the Springer Nature publishing group but is shown as the BMC family of journals here. Complete data included in S1 Data Table A. Colors are used to assist visual separation of blocks and do not represent data types or categories.

**Fig 3. Number of publications by year: (A) single-factor NHANES analyses identified in this review (B) total publications by year identified by a PubMed search for "biobank".** Both data series show publication counts for 2024 to 9 October only. The data underlying these figures are additionally reported in S1 Data Table D.

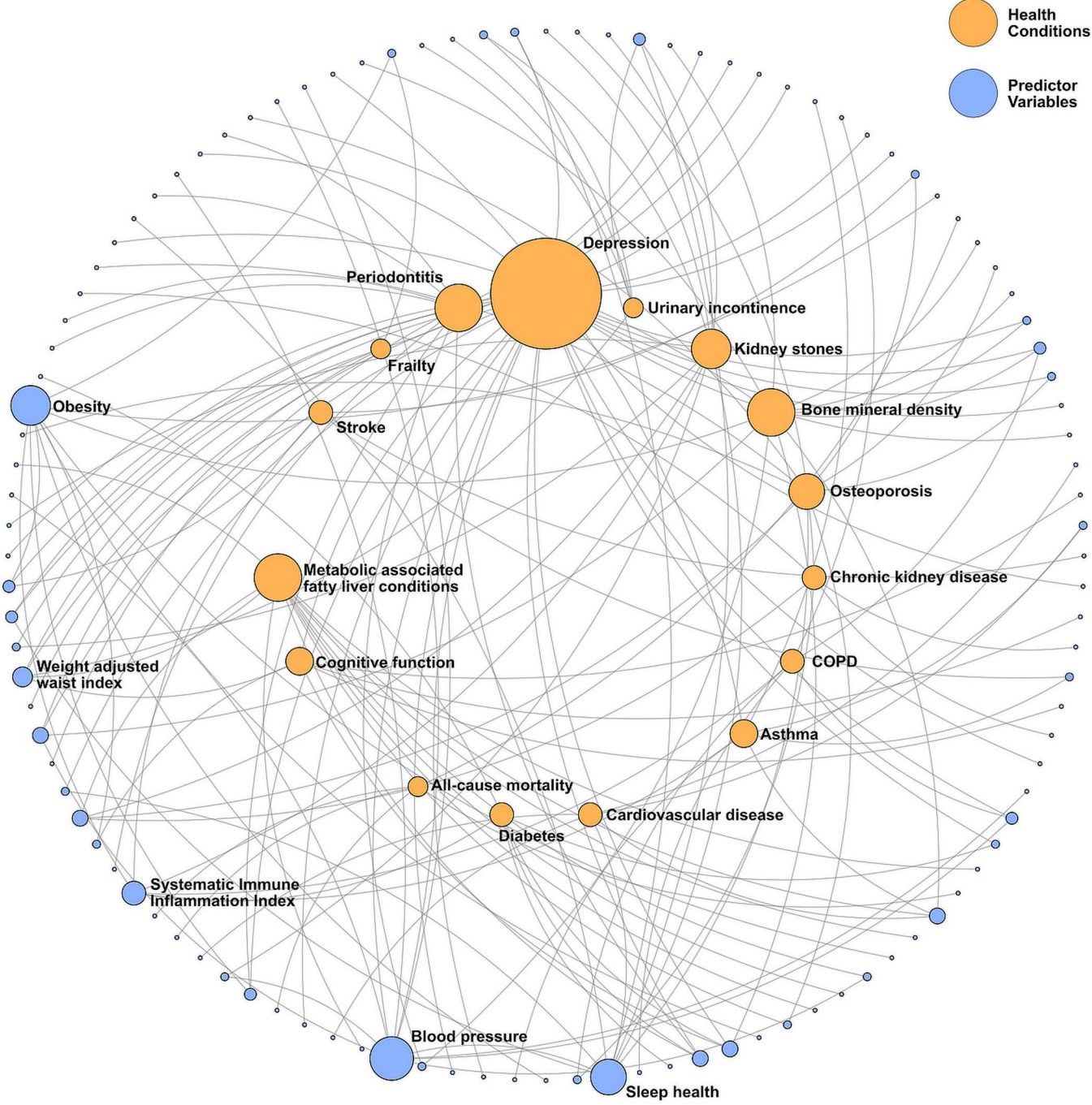

**Fig 4. Network analysis of health conditions and their associations with predictor variables from NHANES published research focusing on single-factor relationships.** Outer shell nodes are predictor variables, inner shell nodes are health conditions with four or more associated predictor variables and edges represent a paper in this systematic review describing a statistically significant single-factor relationship. Diameters of nodes are proportional to the number of papers. Only nodes with more than four edges are labeled. The data underlying these figures are taken from the pivot tables presented in S1 Data Tables B and C.

hypothesis. Depression was analyzed more frequently than any other condition, with 28 individual studies, all but 4 of which were published in 2023 or 2024. The associated independent variables are shown in Table 2 as a case study. The individual studies did not employ false discovery correction, but taken together, represent multiple hypotheses. To compensate for this, False Discovery Rate (FDR) correction using Benjamini-Yekutieli was then applied to these studies using a count of 28 potential relevant hypotheses. Of the 28 statistically significant associations, less than half (13) remained statistically significant after FDR correction.

## 2.3  Selective analysis: Using subsets of available data

As a case study in selection of subsets of data, we investigated the relationship between the systematic immune-inflammation index (SII) and different indicators (Table 3). We compared the data analyzed with the data actually available at the time of each paper's publication (excluding 2021–2023 data, as this was only published by NHANES in September 2024). Notably, the relationship between SII and diabetes was analyzed twice, once for the general US population

**Table 2.  Summary of papers identifying associations with depression. All manuscripts use NHANES data, 2014-2024. Articles listed by date of publication.**

| PMID | Year Published | Predictor variable associated with depression | Reported or derived p-value | FDR p-value |
|---|---|---|---|---|
| 39277033 | 2024 | Serum neurofilament light chains | 0.039 | 0.161 |
| 39355374 | 2024 | Blood pressure [a] | 0.044 | 0.173 |
| 38869164 | 2024 | Life's Essential 8 | <0.001 | <0.001 |
| 39121312 | 2024 | Triglyceride-glucose index | <0.001 | 0.004 |
| 38970091 | 2024 | Adherence to Mediterranean diet | 0.039 | 0.161 |
| 38355455 | 2024 | Geriatric nutritional risk index | <0.001 | <0.001 |
| 38642902 | 2024 | Overactive bladder | <0.001 | <0.001 |
| 39029685 | 2024 | Muscle mass | 0.008 | 0.059 |
| 38710330 | 2024 | Dietary vitamin C intake | <0.001 | <0.001 |
| 38944294 | 2024 | Immune-inflammation-based prognostic index | <0.001 | <0.001 |
| 38579547 | 2024 | Physical activity | 0.032 | 0.153 |
| 39044344 | 2024 | Lipid accumulation products | 0.031 | 0.153 |
| 38154580 | 2024 | Ethylene oxide levels | 0.021 | 0.133 |
| 38220117 | 2024 | Weight-Adjusted Waist Index | 0.008 | 0.059 |
| 38910137 | 2024 | Blood cadmium | <0.001 | <0.001 |
| 37838268 | 2024 | Non-HDL cholesterol to HDL cholesterol ratio (NHHR) | 0.031 | 0.153 |
| 37775007 | 2023 | Per- and polyfluoroalkyl substances (PFAS) | <0.001 | <0.001 |
| 39227001 | 2023 | Non-high-density lipoprotein cholesterol | 0.023 | 0.133 |
| 37474898 | 2023 | Dietary anthocyanidins intake | 0.020 | 0.133 |
| 37236270 | 2023 | Serum α-Klotho | 0.022 | 0.133 |
| 37447273 | 2023 | Serum Vitamin D | 0.032 | 0.153 |
| 36462606 | 2023 | Dietary Inflammatory Index (DII) | <0.001 | <0.001 |
| 37340352 | 2023 | Serum albumin | 0.037 | 0.161 |
| 36265728 | 2023 | Organophosphorus pesticide exposure | 0.040 | 0.161 |
| 36008859 | 2022 | Physical activity | <0.001 | <0.001 |
| 33292309 [17] | 2020 | Ovariectomy-reduced hormones | <0.001 | <0.001 |
| 28395506 | 2017 | Inflammatory bowel disease | 0.002 | 0.017 |
| 24636212 | 2014 | Poor dental health | <0.001 | <0.001 |

[a]Study separated systolic and diastolic blood pressure and reported a *p*-value 0.04 for diastolic and 0.09 for systolic blood pressure.

**Table 3. Summary of papers analyzing SII. All manuscripts use NHANES data, 2014–2024. Articles listed by date of publication.**

| PMID | Year published | Health conditions associated with SII | Years analyzed in paper | Years available (NHANES table/var code) |
|---|---|---|---|---|
| 39052624 | 2024 | Liver injury | 2017–2020 | (ALT): 1999–2020 (LBXSATSI) |
| | | | | (AST): 1999–2020 (LBXSASSI) |
| | | | | (GGT): 1999–2020 (LBXSGTSI) |
| | | | | (ALP): 2003–2020 (LBXSAPSI) |
| 39312354 | 2024 | Hearing loss | 2005–2018 | 1999–2020 (AUX) |
| 38974989 | 2024 | Diabetes | 2003–2018 | 1999–2020 (DIQ) |
| 38822015 | 2024 | Overactive bladder | 2005–2018 | 2005–2020 (KIQ480/KIQ044) |
| 39069464 | 2024 | Obesity | 2011–2018 | 1999–2020 (BMXBMI/ BMXWAIST) |
| 39076553 | 2024 | Stroke | 1999–2020 | 1999–2020 (MCQ160f/F) |
| 38988356 | 2024 | Asthma | 2009–2018 | 2001–2020 (MCQ010/25/35/40/50) |
| 39239395 | 2024 | Serum neurofilament light chain | 2013–2014 | 2013–2014 (SSSNFL) |
| 39267962 | 2024 | Atopic dermatitis | 2001–2006 | 2001–2006 (DED061/AGQ180) |
| 36904176 | 2023 | Hyperlipidemia | 2015–2020 | 1999–2020 (LAB13/TCHOL) |
| 37883981 | 2023 | Serum ferritin levels | 2015–2018 | 1999-2002, 2005–2010, 2015–2020 (LBXFER) |
| 38027115 | 2023 | Diabetes | 2017–2020 | 1999–2020 |
| 38076255 | 2023 | Chronic obstructive pulmonary disease (COPD) | 2013–2020 | 2011–2020 (MCQ160g/o) |
| 36895572 | 2023 | Kidney stones | 2007–2018 | 2007–2020 |

between 2003 and 2018 (PMID 38974989), and once for adults surveyed between 2017 and 2020 (PMID 38027115). In fact, diabetes data were available from 1999 to 2020. For the 14 papers analyzing SII, 4 used the full data made available by NHANES, and 10 were selective in the data analyzed.

The overall distribution of analyzed years is shown in Fig 5. The median number of years of NHANES data analyzed by manuscripts reviewed in this meta-analysis was 4, i.e., incorporating two biennial surveys. The most common 'window' for analysis was 2007–2018 (28 papers) followed by 2005–2018 (23 papers).

## 3. Discussion

The last 3 years have seen a rapid increase in the number of publications analyzing single-factor associations between predictors (independent variables) and various health conditions (dependent variables) using the NHANES AI-ready health and nutrition dataset. An average of 4 papers per annum were published between 2014 and 2021, increasing to 33, 82, and 190 in 2022, 2023, and 2024 (to October 9 only) respectively, with recent formulaic manuscripts targeting a relatively small number of indexed journals with moderate impact factor (averaging 3.6). The rapid acceleration in the publication rate was also accompanied by a shift in the country of origin of publication. While this may be a product of different incentives in China leading to misconduct [18,19], it may also reflect the ability of small networks or groups of individuals to exploit AI-ready datasets for end-to-end generation of very large numbers of manuscripts.

We view this increase in single-factor associative research as increasing the risks of misleading findings being introduced to the body of scientific literature, for a number of reasons. The first is essentially a problem of study design, where clear multifactorial health issues are analyzed as single-factor studies. For example, depression, cardiovascular disease, and cognitive function are all extensively described in the literature as multifactorial in nature [20–22], and yet were investigated in the papers included in this systematic review using essentially single-factor approaches focusing on one predictor, albeit typically including covariates such as sex, age, smoking history, and educational level. This creates risks of incomplete information, bias, and lack of understanding of interactions between predictors of health conditions.

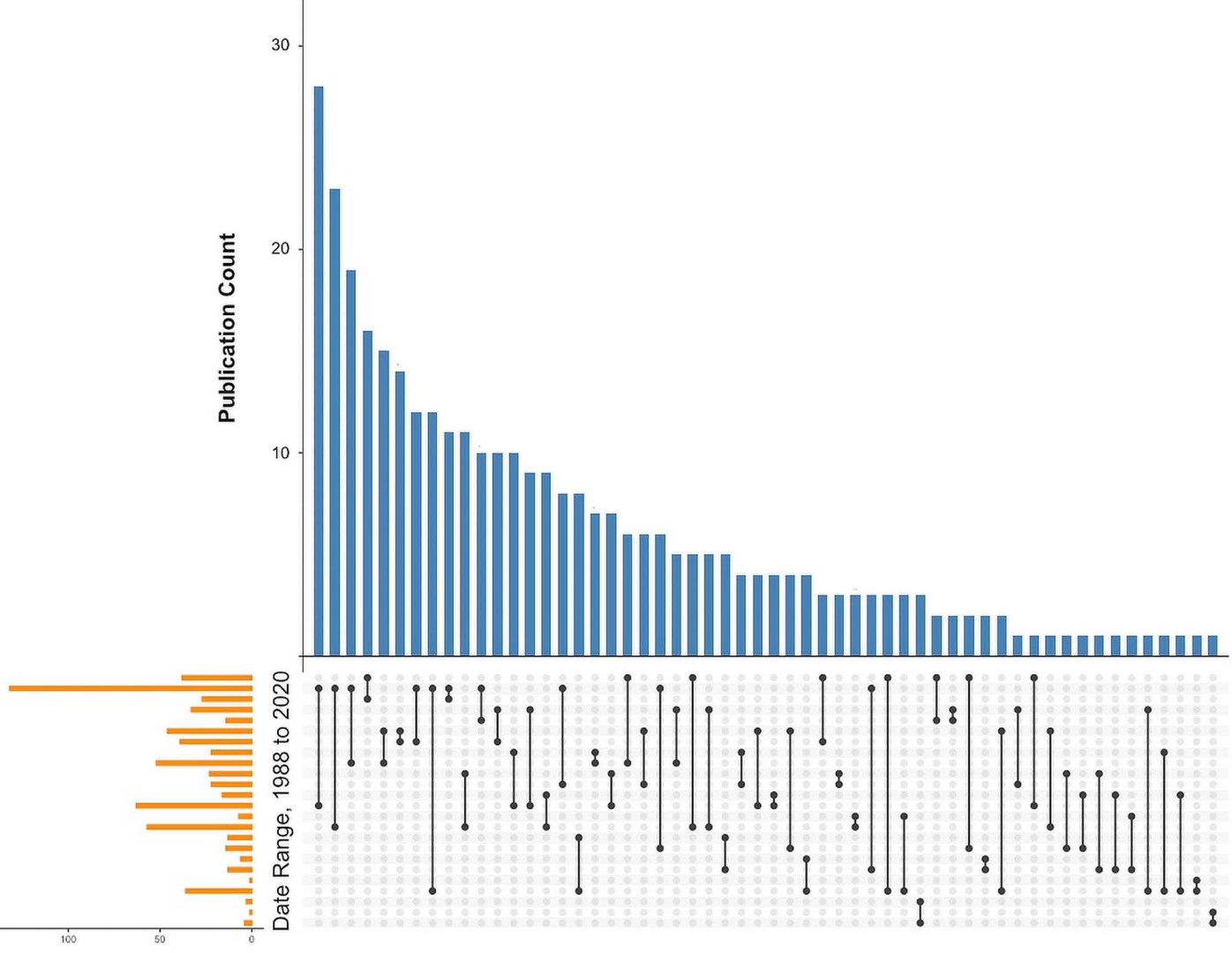

**Fig 5. UpSet plot illustrating selections of annual data taken from NHANES that were employed for the 341 publications analyzed in this work.**
The data are taken from S1 Data Table A, using the start dates to the end dates of subsets of NHANES data analyzed by each manuscript.

Single-factor analyses can also cause problems in the other direction, when a single predictor variable is linked to many health conditions, for , obesity or systematic inflammation. This has previously been demonstrated in UK Biobank data, where latent systemic connectivity was found in common biomarkers across a wide range of health conditions [9]. Best practice is to conduct research in a comprehensive manner, or run the risk of research not being translational [23], with real costs to healthcare systems and patients [24,25].

In terms of productivity, the formulaic nature of the studies reviewed here does have advantages when it comes to producing rapid responses to timely issues; the large majority used the same odds ratios methodology, but speed of production is in our view less relevant when analyzing NHANES data. Indeed, in some cases, the formulaic nature and high productivity seem motivated more by the desire to produce as many manuscripts as possible. For example, one research group simultaneously submitted two analyses of linkages with body shape index (one to abdominal aortic calcification, and one to cognitive impairment), to two different journals, both of which were accepted (PMIDs 39377074 and 38840158).

As well as failing to identify the multiplicity of features that may contribute to a multifactorial health condition, isolated analyses also lack false-discovery correction. The statistical outputs from one research group investigating 28 features and applying FDR correction could be completely different to the outputs from 28 groups investigating features individually. This can lead to differences in findings from the same data and conflicting literature evidence. The avoidance—deliberate or otherwise—of conducting appropriate and holistic investigation of all relevant factors will increase the number of false discoveries (Type I errors). As shown here for depression, under Benjamini–Yekutieli FDR correction 15 out of 28 results would lose statistical significance if analyzed together. FDR correction of data from AI-ready databases will be contentious because determining the number of potential hypotheses is not trivial. FDR correction may not even be necessary in validation work, whereby a feature that has been found significant in one dataset is tested for validation in another dataset. Given that the manuscripts reviewed here were published over a very short period of time using a single data source (NHANES), we view applying FDR correction to these 'multiple hypotheses' as a reasonable approach to highlight the problem.

Furthermore, the availability of AI-ready datasets that can be dredged for relationships creates additional problems of inappropriate subsetting of data, without justification. Here, we provide examples of NHANES data being extracted selectively. For example, while data was available from 1999 to 2020 in NHANES for diabetes and SII, two papers analyzing this relationship selected a period of 2003–2018 and 2017–2020 (PMIDs 38027115 and 38974989). Other forms of selective post-hoc hypotheses may have been employed, given that many of the manuscripts in this meta-analysis discussed subsets, for example, particular age ranges, or sex variations. It is not possible to know which are examples of hypothesizing after the results are known (HARKing) [10,11], but the lack of disclosure and justification of choosing subsets is clearly a shortfall versus best practice.

Two main issues with the work presented here should be highlighted. First, the search strategy was specific and narrow, for reproducibility and to focus on the particular problem of single-factor research, but may understate the numbers of NHANES-related manuscripts that have been produced by data dredging. The emphasis here is to identify the problem and make recommendations for stakeholders, rather than to provide a definitive list of potentially problematic works, but it should be noted that the total number of NHANES manuscripts identified by a simple search of the PubMed database (searching for NHANES at https://pubmed.ncbi.nlm.nih.gov/, accessed 3 March 2025) increased from 4,926 in 2023– 7,876 in 2024. This is suggestive that the scale of the problem could reach thousands of manuscripts, rather than hundreds. Such estimates would be concordant with the trend previously identified in the field of mendelian randomization, another area which has seen a tsunami of formulaic manuscripts [26]. Second, our analysis is not intended to criticize individual manuscripts or attribute them to paper mills; even after FDR correction, many single-factor associations were still statistically significant and of interest to the scientific community. Nonetheless, these limitations do not detract from the wider point, that holistic research is a better way to deal with the challenges posed by big data than the manufacture of very large numbers of single-factor manuscripts. These findings also provide a case study of strategies that may be employed by paper mills. In particular, we have shown here how AI-assisted workflows can accelerate manuscript production using direct API-level access to derive data and results, potentially also using generative AI with the textual aspects of manuscript preparation. It is also plausible that the use of high-quality legitimate datasets such as NHANES is part of a strategy that may assist paper mills in avoiding detection, and which may result in manufactured publications outnumbering legitimate publications in certain data-driven fields. These are all key questions and issues that have recently been highlighted by the United2Act Research Group and others [1,27,28], as well as being part of the wider debate around AI usage in research by bodies including the European Commission [29].

## Recommendations

While identifying these problems is important, proposing recommendations to guard against these issues is equally vital. While organizations such as Committee on Publication Ethics (COPE) provide support and guidance to editors,

publishers, and researchers [30], often peer reviewers will be the last, and therefore the most important, line of defence against poor practices [31], including those that may be adopted by paper mills. While there are challenges in common with those posed by manufactured manuscripts using falsified data [32,33], data dredging from open access datasets creates different issues (as the statistics and data may—in isolation—be sound), and will require different countervailing measures. Here, we set out a number of suggestions for journals, reviewers and policymakers that may help to guard against poor statistical analysis of AI-ready large datasets.

1. While multifactorial research brings its own challenges, researchers should recognize that while a single-factor analysis may produce statistically significant results, prominent acknowledgment should be made of the potential for false discoveries or conflation of specific and general indicators [34]. For editors and reviewers, we see single-factor analysis of conditions known to be complex and multifactorial as a 'red flag' for potentially problematic research.

2. Data providers should use API keys and application numbers to prevent data dredging and hypothesis drift. This is an approach employed by the UK Biobank, for example. Providers should include in their conditions that publications referencing their data include an auditable account number. This step would be consistent with Bridge2AI recommendations that biomedical AI datasets should have ethical constraints, such as proper licensing and distribution with barriers against misuse [3]. It would also mitigate the risks of data dredging by paper mills, in-line with similar approaches such as pre-registration of studies or models such as OpenSafely employed by the UK's National Health Service [35].

3. Publishers should emphasize the importance of desk rejection to reduce the burden on peer reviewers. Stender and colleagues (2024) suggested the use of a template for efficient rejection of papers using formulaic approaches to two-sample mendelian randomization, which we have adapted here as an example. '*A general comment regarding this type of NHANES study is that they have become very easy to do, owing to ready-to-use R and Python packages. The result is an avalanche of NHANES studies being produced and submitted for publication. Many of these stem from research papermills that churn out papers using identical methods and analytical pipelines, only changing the dependent and independent variable between each paper. These papers add little - if anything - of scientific value, and increase the risks of misleading results being introduced to the literature. I therefore recommend that the paper be rejected*'.

4. Publishers should also give consideration to appointing dedicated statistical reviewers, a practice already being adopted by some journals [36]. They should actively seek to identify patterns, and be more proactive in rejecting publications that appear formulaic, as seen here from titles to methodology, where data-driven approaches are used to mass-produce manuscripts from biobanks and similar data sources. When dubious patterns are identified, or there is evidence of subset analysis without justification, publishers should contact authors and ask them to clarify their process, provide updated statistical results, or risk having their papers removed from open access. Publishers should ensure that comments or replies to articles are published promptly without fear or favor, and ensure that manuscripts are withdrawn where appropriate.

5. The scientific community at large can also contribute through proactive follow-up commentary using post-publication fora such as PubPeer [37]. It should be recognized, however, that even when issues are identified, they are not always acted on; a recent study found that in the majority of cases, even where a serious issue was identified (including inter alia honest errors, methodological flaws, publishing fraud or manipulation), no correction was issued by the journal [38]. Nor is it obvious what the consequences are for journals of declining to engage with post-publication appraisal of published work.

In conclusion, this analysis has revealed a concerning trend toward unspecific and uncorrected use of the NHANES health and nutrition dataset via single-factor analyses of the associations between health indicators and multifactorial health conditions. The lack of false-discovery correction results in increased Type I errors, and the selective use of data

subsets without clear justification invites post-hoc hypothesizing (HARKing), which may challenge the reproducibility and reliability of such studies. There is also a risk that the potential for AI-assisted mass production of formulaic research papers may be targeted by paper mills. These issues call for improvement, and addressing them will require action from researchers, journals, and peer reviewers. We recommend a set of easily implemented measures that aim to reduce data dredging, mandating full dataset analyses unless subsets are justified, and integrating dedicated statistical reviewers in the peer-review process. For the longer term, given the growth of big data and AI-ready resources in health and related fields, as well as the potential for paper mills to exploit such assets, we believe that a wider debate around data standards and principles is inevitable.

## 4. Materials and methods

### 4.1 Systematic identification of articles: Bibliometric analysis

A systematic approach was used for the identification of a corpus of NHANES-derived research studies, focused on single-factor research associating a predictor to a health condition. Searches were performed in the PubMed and Scopus databases. The following terms were required in the search strategy, restricted to the title, with alternatives as shown using Boolean operators: NHANES AND (correlation* OR association*) AND (cross-sectional OR population). For all articles identified using this search strategy, titles, and abstracts were screened for eligibility. In this work, the eligibility criteria for inclusion in the systematic analysis were as follows: (a) articles published between January 1st 2014 and October 9, 2024, (b) primary research studies derived directly from NHANES data, and (c) focusing on single-factor research that proposed statistically significant associations between markers or clinical predictors and health conditions/diseases. Studies conducting multifactorial analyses were excluded. Additionally, articles in non-Roman characters, review articles, letters to editors, and corrections were excluded.

Following the PRISMA guidelines the above search and eligibility steps were carried out by two researchers (T.S. and A.E.A.), with differences in identified articles reviewed by a third author (M.S.) for inclusion/exclusion. The relevant articles were then read in full, and data were extracted for meta-analysis (Table 4). Bibliometric trends were then investigated over time as well as by journal.

### 4.2 Statistical issues arising from single-factor study design

**4.2.1 Identification of multifactorial conditions analyzed as single-factor problems.** To investigate whether multifactorial relationships existed in the cohort of manuscripts identified, a network analysis was applied to summarize the relationships between health conditions and the predictor variables that had been associated with them. This network was then visualized, with two classes of nodes (health conditions and associated predictor variables) connected by edges (papers proposing a pairwise statistically significant association). Where nodes had multiple connections, this

**Table 4. Extracted data fields for meta-analysis.**

| Data field | Comments |
|---|---|
| Included | Yes/No |
| Reason for Exclusion (if applicable) | Review articles, letters to editors/ replies, and corrections were excluded |
| Independent Variable | Feature associated with the health condition |
| Dependent Variable | The health condition being investigated |
| Cohort Population | Adults, Children, Male, Female, General Population |
| Cohort Start/ End Date | Range of years analyzed |
| Journal | Where the manuscript was published |
| Affiliation | Affiliation of the corresponding author |

was considered to be evidence of a multifactorial relationship that could have been captured by appropriate analysis of NHANES data. Network analysis was conducted in Python using the networkx library.

**4.2.2 False discovery correction.** Another issue that arises when investigating multifactorial health conditions as single-factor problems is that by focusing on single factors only, manuscripts will lack the false discovery corrections that would be employed if the complete range of relationships were to be investigated. FDR correction for all studies identified in the systematic review was beyond the scope of this work, so FDR correction was applied to the most commonly identified health condition in this work as a case study, allowing for comparison of the strength of relationships across predictor variables, as well as identifying those that might not reach statistical significance when corrected for multiple hypothesis testing. Benjamini–Yekutieli correction was used given the potential for dependencies and correlations within NHANES [39], with $n_{hypotheses}$ set to the number of single-factor associations proposed in the examined corpus of publications. This was a smaller value for $n_{hypotheses}$ than the total number of independent variables present across NHANES, given that some independent variables would have had no relevance to the condition; identifying the true number of potential hypotheses that could have been tested was not deemed possible. Where a $p$-value for a finding was provided in a research paper, this was taken as presented and FDR correction applied. Where an odds ratio (OR) and confidence interval (CI) was given, we log-transformed these values and then calculated the standard error as $\left(log\left(Upper\ CI\right) - log\left(Lower\ CI\right)\right)\ /\ \left(2\ x\ Z_{0.975}\right)$. The $z$-score was then computed as the ratio of the log-transformed odds ratio to the standard error, and the $p$-value was calculated based on the $z$-score using the cumulative distribution function of the standard normal distribution. FDR correction was then applied to this derived $p$-value. This allowed the identification of findings that were statistically significant on an individual basis, but would not have been found to be statistically significant had a more wide-ranging and comprehensive investigation been performed, testing multiple independent variables. FDR correction was conducted in Python using the scipy.stats library.

### 4.3 Selective analysis

While NHANES does adjust the scope of its questionnaires from time to time, and not all data are available across all biennial surveys, many series are collected over an extended period of time. In order to investigate whether there was evidence of data dredging of limited subsets of data, we compared the dates analyzed in a subset of manuscripts with the dates for which NHANES data was available, to identify any mismatches. Where authors used a subset of a time-series without any justification, we considered this to be an example of selective analysis.

### Supporting information

**S1 Data. Metadata and other supporting materials for the the publications reviewed in the systematic analysis.** (XLSX)

### Acknowledgments

The authors additionally wish to acknowledge the wider support of the AIBIO-UK network, [https://aibio.ac.uk/] and also the AIRBDS working group formed as part of AIBIO-UK.

### Author contributions

**Conceptualization:** Charlie Harrison, Matt Spick.

**Data curation:** Tulsi Suchak, Anietie E. Aliu.

**Formal analysis:** Matt Spick.

**Investigation:** Tulsi Suchak, Anietie E. Aliu, Matt Spick.

**Methodology:** Charlie Harrison, Matt Spick.

**Project administration:** Nophar Geifman, Matt Spick.

**Resources:** Nophar Geifman.

**Software:** Tulsi Suchak.

**Supervision:** Reyer Zwiggelaar, Matt Spick.

**Validation:** Anietie E. Aliu, Reyer Zwiggelaar.

**Visualization:** Tulsi Suchak.

**Writing – original draft:** Tulsi Suchak, Anietie E. Aliu.

**Writing – review & editing:** Charlie Harrison, Reyer Zwiggelaar, Nophar Geifman, Matt Spick.

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
