## [Editor Report · Decision Letter 0]

18 Jan 2025

Dear Dr Spick, 

Thank you for submitting your manuscript entitled "Analysis of NHANES-based research identifies risks of data dredging, false discoveries and misleading findings" for consideration as a Research Article by PLOS Biology.

Your manuscript has now been evaluated by the PLOS Biology editorial staff, as well as by an academic editor with relevant expertise, and I'm writing to let you know that we would like to send your submission out for external peer review.

Once your full submission is complete, your paper will undergo a series of checks in preparation for peer review. After your manuscript has passed the checks it will be sent out for review. To provide the metadata for your submission, please Login to Editorial Manager (https://www.editorialmanager.com/pbiology) within two working days, i.e. by Jan 21 2025 11:59PM.

Kind regards,

Roli Roberts

Roland Roberts, PhD

Senior Editor

PLOS Biology

rroberts@plos.org

---

## [Decision Letter · Decision Letter 1]

26 Feb 2025

Dear Dr Spick,

Thank you for your patience while your manuscript "Analysis of NHANES-based research identifies risks of data dredging, false discoveries and misleading findings" went through peer-review at PLOS Biology. Your manuscript has now been evaluated by the PLOS Biology editors, an Academic Editor with relevant expertise, and by four independent reviewers.

You'll see that reviewer #1 thinks that this study is important, but suggests that you use PMIDs instead of citing the questionable papers (another reviewer strongly agreed with this during cross-commenting), requests several points of discussion, and suggests a standardised reviewer template. Reviewer #2 is also very positive, and just has two points for discussion. Reviewer #3 is positive, but has a number of issues that will need to be addressed in order to convince. One is that she doesn’t think that the problems will necessarily be solved by multi-factorial analysis, and the other is that the patterns seen can emerge from research assessment structures in countries like China, without necessarily involving paper mills (this may be possible to address by analysis?). She also thinks that you should dwell more on solutions (cross-validation, pre-registration, guidelines for journal editors). Reviewer #4 is also very positive, and suggests a tweak to your search criteria, suggests that you cite his editorial (other reviewers agreed), and recommends mentioning the analogous concerns raised with Mendelian Randomisation studies.

In light of the reviews, which you will find at the end of this email, we are pleased to offer you the opportunity to address the comments from the reviewers in a revision that we anticipate should not take you very long. We will then assess your revised manuscript and your response to the reviewers' comments with our Academic Editor aiming to avoid further rounds of peer-review, although we might need to consult with the reviewers, depending on the nature of the revisions.

**IMPORTANT - SUBMITTING YOUR REVISION**

*Resubmission Checklist*

*Published Peer Review*

*PLOS Data Policy*

*Blot and Gel Data Policy*

Sincerely,

Roli Roberts

Roland Roberts, PhD

Senior Editor

PLOS Biology

rroberts@plos.org

REVIEWERS' COMMENTS:

Reviewer #1:

[identifies herself as Jennifer A. Byrne]

This manuscript describes an important problem, namely the possible exploitation of NHANES-based research data for the scaled production of low value research manuscripts. The manuscript describes a steep recent rise in the number of articles published using NHANES data. It is important for these results to be communicated rapidly, as it would seem that the numbers of such manuscripts could continue to rise, possibly overwhelming editorial and peer review processes at some journals. 

I should specify that my expertise lies outside the types of statistical analyses performed, so I wasn't able to critically evaluate all the data.

Major issues:

Page 6: I strongly suggest to not cite the studies listed in Tables 2 and 3, but to instead use PubMed ID's within Tables and text. Paper mills are likely to value to citations to their work, so citing many problematic papers inadvertently supports this model. 

Page 10: "The challenges analysed here are different to those for manufactured manuscripts using falsified data"- this warrants further discussion. Arguably, some challenges are the same, ie fabricated and genuine yet low value/derivative manuscripts can both report unreliable results, both classes of manuscripts waste editor and peer reviewer time. It would be worth explaining the similarities and differences between manuscripts generated from AI-ready datasets versus those that are entirely manufactured from fabricated data, so that any specific issues are clearer.

Page 10: "dedicated statistical reviewers"- many journals would indeed benefit from statistics reviewers, however, their time could be easily consumed by low value manuscripts. The authors then call for better manuscript screening processes, which are essential for expert reviewers to focus attention on quality submissions. This point should be expanded, eg by referencing recent publications https://doi.org/10.1210/clinem/dgaf036 (this very recent editorial mentions similar manuscripts to those described here), also https://lipidworld.biomedcentral.com/articles/10.1186/s12944-024-02284-w. More emphasis should be placed on the importance of desk rejections to save editorial and peer reviewer time, as per https://doi.org/10.1210/clinem/dgaf036. Stender and colleagues have proposed a peer reviewer template to recommend rejection of low value 2SMR manuscripts- this could be built upon to provide similar resources for reviewers of NHANES-based manuscripts. This would seem to be a valuable addition.

Minor issues:

Page 1: Manuscript subtitle "Unethical research practices"- could this be "Questionable", "Problematic"? Research ethics doesn't seem to be a focus of the manuscript.

Page 2: The abstract is quite long and includes some details that could be omitted.

Figure 1: The authors could check the n values shown, eg 426-6 does not equal 417.

Figure 1: legend refers to a "systematic review", whereas the text refers to "meta-analysis" (lines 124, 221, 307).

Figure 2: The legend does not describe the colours used, some of which are used more than once. Are the colours significant?

Page 5: "chow test" (used twice), "Chi-square test" are not mentioned in the Methods.

Page 5: "biobank" is used as a keyword to show recent increases in data-driven research (Figure 3B), yet most human health biobanks provide biospecimens and associated data, and hence support laboratory research that's not primarily data-driven. The authors could therefore rethink the use of "biobank" as a control keyword.

Figure 3: Colours used are not defined.

Figure 5: Suggest "Publication count" for Y axis, also "publications" in legend.

Page 8: Suggest inserting a paragraph break at line 194- paragraph is currently quite long.

Page 10: "often peer reviewers will be the last"- this is an important point, however the cited reference from 2008 would have been unlikely to specifically refer to paper mills. This sentence could be reworded for clarity.

Page 10: "early warning lists"- it's unclear what's intended here. Some readers will associate this term with the CAS journal early warning lists, eg: https://ewl.fenqubiao.com/#/en/early-warning-journal-list-2024.

Reviewer #2:

[identifies himself as Nikolaus C Netzer MD PhD]

I completely support this manuscript by all means, agree in all points with the authors and think it is extremely important to get the message out. I have a few minor requests based on my own experience with the topic.

1. I know that GB is not a part of the EU anymore, still research strings are tight. I would shortly discuss in the paper the EU AI in research act. Especially since some influential economical experts recently criticize the EU for being to restrictive with AI regulations and that it would therefore harm the economical and scientific progress and success of EU countries. I think this critizism is only money driven and lacks any ethical aspects of the topic (Netzer NC. Artificial intelligence - the Janus-faced tool in our hands. Sleep Breath. 2024 Oct;28(5):1861-1862. doi: 10.1007/s11325-024-03129-7. Epub 2024 Aug 5. PMID: 39098968; PMCID: PMC11449974; https//research and innovationec.europa-eu./research area/industrial research and innovation/ Artificial Intelligence (AI) in Science - European Commission (europa.eu) ).

2. The problem with single parameter correlations is not only that they might deliver false results. I found a MR NHANES manuscript as reviewer, where the AI hallucinated by simply turning the hypothesis around from two single parameters. The first result was NHANES data of depression in subjects correlated with data of sleep apnea and the conclusion of the paper (the AI) was sleep apnea leads to depression. That's scientifically not proven (most patients have daytime sleepiness, but do not develop a depression from that) but possible. Then I got the same manuscript again to review with the authors in reversed order, turning the conclusion around (same correlation), that depression leads to sleep apnea. Physiologically and pathophysiologically total nonsense. I described that problem in the above listed editorial in Sleep and Breathing.

Maybe you want to build that problem with reversed single parameter correlation into your arguments respectively discussion or introduction. 

Reviewer #3:

[identifies herself as Dorothy Bishop]

Summary

After a move to encourage researchers to adopt open data, it is becoming clear that unintended negative consequence is data dredging of open datasets that leads to pointless analyses being published. The authors demonstrate this phenomenon with the NHANES dataset, showing that there has been a rapid growth in papers reporting associations between single predictors and single outcomes. A high proportion of the recent paper originate from China. The authors argue that this research is flawed because it does not take into account the multifactorial origins of most health conditions, and that AI-ready datasets may be misused by paper mills. They conclude with some recommendations for overcoming this situation.

Overall evaluation

The authors have conducted a systematic analysis of papers from the NHANES dataset, and provide useful empirical evidence to support the view that single-factor analyses have skyrocketed in recent years. This is useful but I see two aspects that I am not convinced by and that I'd like to see addressed: (a) the need for multifactor analyses and (b) the involvement of paper mills. In addition, I think the recommendations could be a lot stronger.

Multifactor analysis

While I agree that single factor accounts of most health conditions are implausible, I'm not sure the problem identified here would be solved by researchers adopting multi-factorial analyses. It could be argued that it would just make the problem worse, given that the number of potential associations increases multiplicatively as more factors are included in the analysis. In other words, unless analyses are pre-registered and theoretically or empirically motivated, encouraging authors to just include yet more variables in the analysis will make matters worse rather than better, because the chance of false positive findings increases dramatically. So I think the focus on single vs multiple factors is a bit misleading here.

Involvement of paper mills

Paper mills were mentioned at several points during the article, but no evidence was provided for their involvement in the explosion of single factor papers. Given that these mostly originate in China, where the incentives for publishing in Western journals have changed dramatically, it's not clear that we are necessarily seeing the result of paper mill activities. See this article that discusses how incentives can lead to misconduct but which does not feature paper mills as a mechanism:

Zhang, X., & Wang, P. (2024). Research misconduct in China: Towards an institutional analysis. Research Ethics, 17470161241247720. https://doi.org/10.1177/17470161241247720

Diagnosing paper mills is not a straightforward business, and I can see it would be beyond the scope of this article to attempt to do such an analysis. Nevertheless, it might be feasible to look at two indicators: the geographical diversity of authors, and the use of non-institutional emails, both of which have been identified as red flags for paper mills: Van Noorden, R. (2023). How big is science's fake-paper problem? Nature, 623(7987), 466-467. https://doi.org/10.1038/d41586-023-03464-x. Alternatively, the paper could be rewritten to make it clear that the statements about paper mills are conjecture and not supported by evidence.

Recommendations for action

An additional recommendation would be to provide access to just half of the dataset, which could be used for exploratory analysis. The researcher would then have to show that the results replicated in the other half - a kind of cross-validation sample.

Also, the OpenSafely model used for accessing NHS data provides good protection against data dredging: Nab, L., Schaffer, A. L., Hulme, W., DeVito, N. J., Dillingham, I., Wiedemann, M., Andrews, C. D., Curtis, H., Fisher, L., Green, A., Massey, J., Walters, C. E., Higgins, R., Cunningham, C., Morley, J., Mehrkar, A., Hart, L., Davy, S., Evans, D., … Goldacre, B. (2024). OpenSAFELY: A platform for analysing electronic health records designed for reproducible research. Pharmacoepidemiology and Drug Safety, 33(6), e5815. https://doi.org/10.1002/pds.5815

Requiring authors to pre-register their research hypothesis and making data access contingent on this sounds similar to the option 2 discussed by the authors, but I was not clear how far that required explicit specification of a hypothesis.

In genetics, where data dredging led to a decade or so of non-replicable candidate gene studies, it became mandatory to replicate the results in a fresh sample. This is similar to the cross-validation idea above, but may be more suited to a case where the sample size is small.

The kind of rapid growth in a methodology described here is reminiscent of what has been reported in other areas: Stender, S., Gellert-Kristensen, H., & Smith, G. D. (2024). Reclaiming mendelian randomization from the deluge of papers and misleading findings. Lipids in Health and Disease, 23(1), 286. https://doi.org/10.1186/s12944-024-02284-w. Those authors had pretty stringent recommendations for journal editors that might be applicable also in this case.

More minor points

p 2 line 26, "Second,..." this is not a sentence; just needs rewriting

line 108. I wasn't sure what was meant by the sentence starting "There was a substantial.."

para using Chow test and chi square test on p 5. The authors have criticised others for reporting p-values after data dredging, but the statistical results reported here have a similar flavour. That is, it is not clear that there was a hypothesis, in which case p-values seem inappropriate. I think the trends are very obvious in any case and that purely descriptive data would be fine here.

p 6, line 146. I was puzzled as to how the FDR correction was applied. This eventually became clear after reading the methods. It would make more sense if the Methods preceded the Results, if the journal allows this. In any case, it seems that the use of FDR correction involves some strong assumptions that may not be justified. One issue is that we don't know how many tests were conducted by went unreported. And as the authors note later on, determining the number of potential hypotheses is not trivial. This is one reason why I think the key issue is not so much whether you have single factor or multi factor analyses, but rather whether you have clearly defined a priori hypotheses. 

p 8 line 172. Again, I don't like referring to a "statistically significant increase" when no hypothesis has been stated. Sorry for being pedantic, but I'd really prefer to just have a descriptive statement here, maybe referring to a N-fold increase.

p 8 line 181. para on single-factor design. This is where I think you need to be careful to not encourage multifactor data dredging. A multifactor analysis should be motivated either by reference to prior literature and/or by theoretical considerations; just throwing lots of variables into the analysis would make matters worse I think.

p 10 ;I suggest new header at end of para 1.

Reviewer #4:

[identifies himself as Stefan Stender]

This study analyzed the rapid growth in publications stemming from single-factor association analyses of the publicly available National Health and Nutrition Examination Survey (NHANES) cohort. A literature search using a set of criteria identified 341 NHANES-derived papers from 2014-2024, each proposing an association between a

predictor and a health condition. The number of papers per year is rising rapidly in recent years. The authors find evidence of data dredging and hypothesizing after results are known ('HARKI'ing'). They highlight the potential of these formulatic NHANES-papers being made (wholly or partly) by AI-methods and/or paper-mills.

Finally, the authors describe a set of best practices to address the concern of this type paper flooding the literature.

The study deals with an important and very timely topic. The paper is well written. The findings and arguments are compelling. 

I have a few minor comments:

1) The search criteria are quite specific: 'NHANES AND (correlation* OR association*) AND (cross-sectional OR population)' 

I realize this is probably required to limit the number of hits. However, I think it's worth reporting the total number of NHANES papers identified just with simple search 'NHANES' by year:

https://pubmed.ncbi.nlm.nih.gov/?term=NHANES&sort=pubdate&timeline=expanded

This is striking. The number per year was rising steadily from 2000-2017, plateauing from 2018 to 2023 at around 4700 per year, and then spiking to N=7818 in 2024! 

After 60 days of 2025 we are already at N=1485. So, the projected number of NHANES papers for 2025 is probably > 10000! In other words, there is evidence that the explosion in NHANES papers is on a much larger scale than the numbers reported from the articles identified by the authors using the narrow search criteria. I think this is worth pointing out.

2) The issue with NHANES papers was to my knowledge first noted in an editorial in 2024: 

https://link.springer.com/article/10.1007/s11325-024-03129-7

This editorial should be cited.

3) A very similar issue is plaguing the field of Mendelian randomization (MR), where we see an explosion in number of papers using publicly available summary data in recent years. This explosion is driven by papers based on two-samples MR (2SMR), where both the exposure and outcome data are based on summary statistics from published GWAS. The rise is entirely driven by papers from China, and there is evidence that some are being produced by paper-mill like factories and using AI. The mass-produced 2SMR has been the focus of several recent editorials and comments: PMID: 39244551, PMID: 39311417, PMID: 39407214. It would be relevant to highlight this analogous situation in the discussion of the present manuscript.

---

## [Editor Report · Decision Letter 2]

19 Mar 2025

Dear Dr Spick,

Thank you for your patience while we considered your revised manuscript "Analysis of NHANES-based research identifies risks of data dredging, false discoveries and misleading findings" for publication as a Research Article at PLOS Biology. This revised version of your manuscript has been evaluated by the PLOS Biology editors and the Academic Editor.

Based on our Academic Editor's assessment of your revision, we are likely to accept this manuscript for publication, provided you satisfactorily address the following data and other policy-related requests.

IMPORTANT - please attend to the following:

a) Please change your Title to something that brings the importance oft the advance more to the fore. We suggest: "Explosion of formulaic research articles, including inappropriate study designs and false discoveries, based on the NHANES US national health database"

b) Please address my Data Policy requests below; specifically, we need you to supply the numerical values underlying Figs 3AB, 4, 5, either as a supplementary data file or as a permanent DOI’d deposition.

c) Please cite the location of the data clearly in all relevant main and supplementary Figure legends, e.g. “The data underlying this Figure can be found in S1 Data” or “The data underlying this Figure can be found in https://zenodo.org/records/XXXXXXXX

d) Please make any custom code available, either as a supplementary file or as part of your data deposition.

We expect to receive your revised manuscript within two weeks. 

*Published Peer Review History*

*Press*

Sincerely,

Roli Roberts

Roland Roberts, PhD

Senior Editor

rroberts@plos.org

PLOS Biology

DATA POLICY:

Regardless of the method selected, please ensure that you provide the individual numerical values that underlie the summary data displayed in the following figure panels as they are essential for readers to assess your analysis and to reproduce it: Figs 3AB, 4, 5. NOTE: the numerical data provided should include all replicates AND the way in which the plotted mean and errors were derived (it should not present only the mean/average values).

CODE POLICY

DATA NOT SHOWN?

---

## [Editor Report · Decision Letter 3]

4 Apr 2025

Dear Matt,

Thank you for the submission of your revised Meta-Research Article "Explosion of formulaic research articles, including inappropriate study designs and false discoveries, based on the NHANES US national health database" for publication in PLOS Biology. On behalf of my colleagues and the Academic Editor, Marcus Munafo, I'm pleased to say that we can in principle accept your manuscript for publication, provided you address any remaining formatting and reporting issues. These will be detailed in an email you should receive within 2-3 business days from our colleagues in the journal operations team; no action is required from you until then. Please note that we will not be able to formally accept your manuscript and schedule it for publication until you have completed any requested changes.

Best wishes,

Roli

Senior Editor

PLOS Biology

rroberts@plos.org